# Structural basis for IL-1α recognition by a modified DNA aptamer that specifically inhibits IL-1α signaling

Xiaoming Ren[1,2], Amy D. Gelinas[3], Ira von Carlowitz[3], Nebojsa Janjic[3] & Anna Marie Pyle[1,2]

IL-1α is an essential cytokine that contributes to inflammatory responses and is implicated in various forms of pathogenesis and cancer. Here we report a naphthyl modified DNA aptamer that specifically binds IL-1α and inhibits its signaling pathway. By solving the crystal structure of the IL-1α/aptamer, we provide a high-resolution structure of this critical cytokine and we reveal its functional interaction interface with high-affinity ligands. The non-helical aptamer, which represents a highly compact nucleic acid structure, contains a wealth of new conformational features, including an unknown form of G-quadruplex. The IL-1α/aptamer interface is composed of unusual polar and hydrophobic elements, along with an elaborate hydrogen bonding network that is mediated by sodium ion. IL-1α uses the same interface to interact with both the aptamer and its cognate receptor IL-1RI, thereby suggesting a novel route to immunomodulatory therapeutics.

[1] Department of Molecular, Cellular, and Developmental Biology, Yale University, 219 Prospect Street, New Haven, CT 06511, USA. [2] Department of Chemistry, Howard Hughes Medical Institute, Yale University, New Haven, CT 06511, USA. [3] SomaLogic, Inc., 2945 Wilderness Place, Boulder, CO 80301, USA. Xiaoming Ren and Amy D. Gelinas contributed equally to this work. Correspondence and requests for materials should be addressed to A.M.P. (email: anna.pyle@yale.edu)

During the past 25 years, the method of systematic evolution of ligands by exponential enrichment (SELEX) has become a powerful tool for the selection of aptamers that bind their target proteins with high affinity[1–3]. However, the limited chemical diversity of nucleic acids has been a constraining factor for developing high-affinity aptamers to many protein targets. With judicious introduction of diverse functional groups at the 5-position of uracil, the repertoire of aptamers has considerably expanded, resulting in a novel class of nucleic acid ligands called Slow Off-rate Modified Aptamers that display exceptionally high affinity and specificity[4, 5]. This advance significantly narrows the diversity gap between nucleic acid ligands and protein ligands (such as antibodies) and greatly improves the success rate for the identification of aptamer ligands against key protein targets, such as cytokines and other signaling molecules.

As a member of the cytokine interleukin 1(IL-1) family[6], IL-1α plays a central role in the regulation of the mammalian immune response[7, 8], with accumulating evidence implicating it in cardiovascular disease, systemic sclerosis, cancer, and other conditions[9–15]. It has been reported that blocking IL-1α with an antibody or interleukin-1 receptor antagonist (IL-1RA) has therapeutic potential for treatment of human inflammatory diseases and cancer[16–18]. Despite its importance, structural studies on IL-1α have been limited, and a high-resolution all-atom structure of the protein has been lacking. Without available data on the surface and side-chains of IL-1α, research on the molecular determinants for receptor binding has progressed slowly.

To address these issues using a new approach, we performed SELEX against IL-1α and successfully isolated a high-affinity ($K_d$ = 7.3 nM) IL-1α-specific modified aptamer designated SL1067, which is a DNA aptamer that contains 2-naphthylmethyl substitutions at the 5-position of deoxyuridine (2Nap-dU). We then evaluated the specificity of SL1067 for IL-1α in receptor competition binding and in cell-based studies where it behaves as a potent inhibitor of IL-1α/IL-1 receptor pathway signaling. To complement these functional studies, we solved a high-resolution crystal structure of the IL-1α/SL1067 complex and deduced the molecular basis for affinity and specificity. The resulting structure provides much-needed insights into the structural features of IL-1α while revealing a strikingly compact aptamer that presents a complex interaction interface with protein. The large hydrophobic naphthyl moiety and the flexible methyl-carboxamide linker enable SL1067 to adopt an unprecedented conformation that facilitates its high-affinity interaction with protein ligand. These studies advance our understanding of cytokine recognition and nucleic acid structure while suggesting new strategies for modulation of the immune system.

## Results

**Selection and characterization of modified aptamer SL1067.** To obtain a modified DNA aptamer with high affinity to human IL-1α, we performed SELEX using recombinant IL-1α protein and a DNA library containing a 40-nucleotide random region in which 5-(2-naphthylcarboxamide)-deoxyuridine (2Nap-dU) was uniformly substituted for dT. Following seven rounds of SELEX and sequencing of the affinity-enriched library, we identified a group of related sequences containing a 17 nucleotide pattern that accounted for 18% of the enriched library. For the predominant

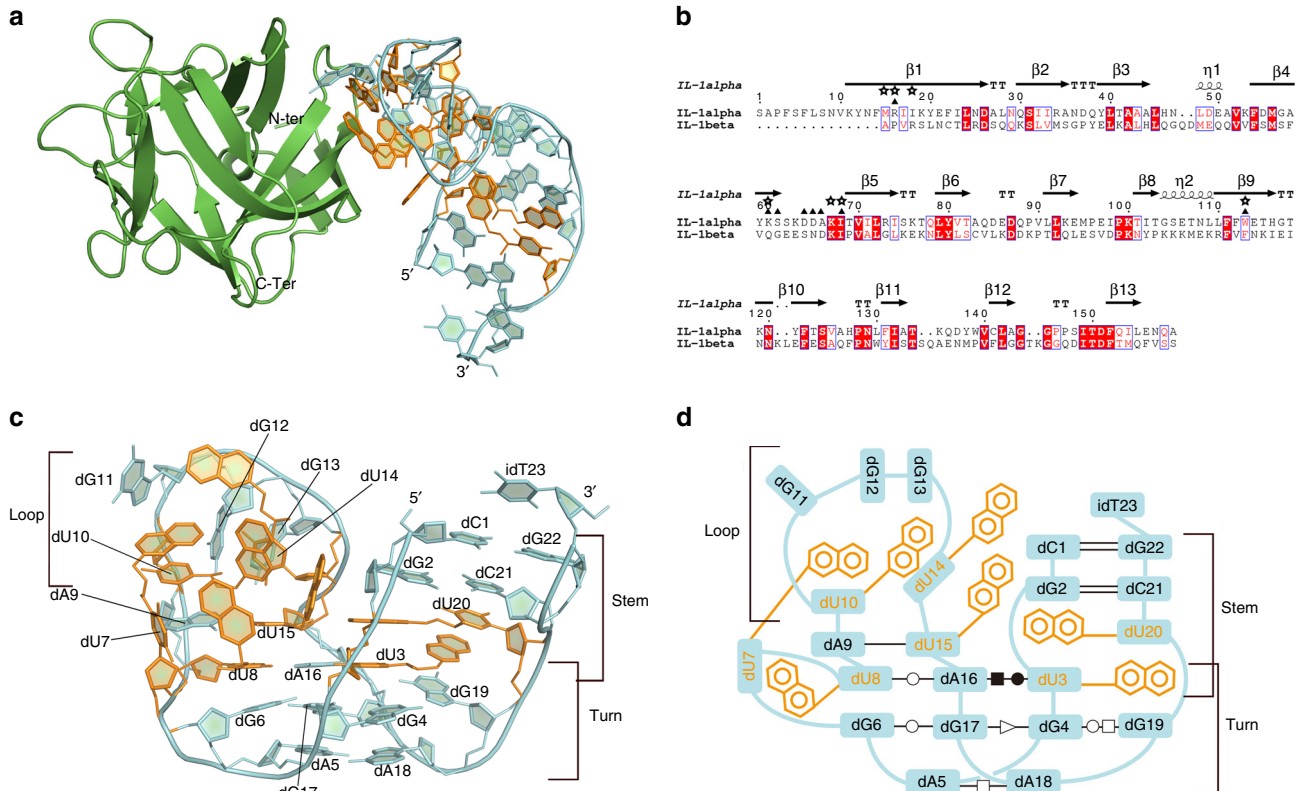

**Fig. 1** Structural and schematic representations of the IL-1α/SL1067 complex and SL1067. **a** Overall structure of the IL-1α/SL1067 complex. IL-1α is colored *green* and SL1067 is colored *cyan* with the 2Nap modified residues colored *orange*. **b** Sequence alignment of IL-1α and IL-1β. IL-1α residues that are involved in the non-polar and polar interactions with SL1067 are denoted with *black stars* and *triangles* separately. **c** Side view of the SL1067 structure. The Nap modified nucleotides are colored *orange*. **d** Schematic representation of SL1067. The three parts of SL1067: stem, turn, and loop are marked. All the base pairs are labeled according to the nomenclature of Leontis et al.[26]

**Table 1 Crystallographic data collection and refinement statistics**

| | |
|---|---|
| *Data collection* | |
| Space group | $P3_1 2 1$ |
| Unit cell parameters: *a, b, c* (Å), *α, β, γ* (°) | 74.79, 74.79, 86.36, 90, 90, 120 |
| Wavelength (Å) | 0.9999 |
| Resolution range (Å) | 64.77–2.10 (2.21–2.10) |
| Unique reflections | 16642 |
| Completeness (%) | 99.5 (100) |
| $R_{merge}$ (%)[a] | 0.092 (0.56) |
| Mean $I/\sigma I$ | 11.2 (3.4) |
| *Refinement* | |
| Resolution range (Å) | 64.77–2.10 (2.21–2.10) |
| $R_{work}$ (%) | 17.76 |
| $R_{free}$ (%) | 21.93 |
| B-factors (Å$^2$) | 38.10 |
| Protein | 37.56 |
| Nucleic acid | 37.74 |
| Metal ions | 37.68 |
| Solvent | 43.45 |
| RMSD bonds (Å) | 0.0072 |
| RMSD bond angles (°) | 1.344 |
| Ramachandran plot | |
| Residues in most favored regions (%) | 97.32 |
| Residues in allowed regions (%) | 2.68 |
| Residues in disallowed regions (%) | 0 |

Values in parenthesis indicate the values for the highest resolution shells
[a]$R_{merge} = \sum_{hkl}\sum_i |I_i(hkl) - \langle I(hkl)\rangle| / \sum_{hkl}\sum_i I_i(hkl)$, where $I_i(hkl)$ is the intensity of the measurement of reflection *hkl* and $\langle I(hkl)\rangle$ is the mean value of $I_i(hkl)$ for all *i* measurements

sequence in this group, we systematically removed nucleotides from the 3′ and 5′ ends to identify the minimal sequence (22mer) that maintained high-affinity binding to IL-1a (SL1067, $K_d = 7.3$ nM) and used this fully truncated ligand for crystallography studies.

**Overall structure of the IL-1α/SL1067 complex.** The structure of the IL-1α/SL1067 complex was solved using molecular replacement and determined at 2.10 Å. The complex has overall dimensions of 58 Å × 32 Å × 30 Å with one IL-1α and one SL1067 arranged side by side within each asymmetric unit (Fig. 1a). The crystallographic data collection and refinement statistics of IL-1α/SL1067 complex are summarized in Table 1. The 1:1 binding ratio of IL-1α to SL1067 was validated by gel filtration chromatography (Supplementary Fig. 1).

To date there are three structures of IL-1α in the Protein Data Bank: a crystal structure that includes only Cα atoms, and two low-resolution nuclear magnetic resonance spectroscopy (NMR) structures[19, 20]. The IL-1α/SL1067 model presented here represents the first complete high-resolution structure of IL-1α, making it possible to now visualize the location and conformation of all protein side chains. In the complex, IL-1α adopts a structure that is identical to that of the free protein in terms of main chain atoms (PDB ID 2ILA, root-mean-square deviation (RMSD) = 0.314 Å), indicating that the structure of IL-1α does not undergo significant conformational changes upon SL1067 binding, consistent with other modified aptamer–protein complexes[21].

As a member of the β-trefoil protein family[22], IL-1α adopts a secondary structure that is composed almost entirely of β-strands with one α-helix. The core of the structure is a six-stranded β-barrel and another six β-strands form three hairpins that serve as the bottom of the barrel. Unlike the other two IL-1 family cytokines, IL-1β and IL-1Ra, which contain 12 β-strands, IL-1α has an additional β-strand at the N-terminus that forms hydrogen

bonds with strand S5 and the long loop L8-9, respectively, further stabilizing the whole protein structure (Fig. 1b).

SL1067 requires only 22 nucleotides for high-affinity binding (with the 3′-inverted dT23 serving as a protective moiety) and is therefore the smallest modified aptamer to be crystallized[23–25]. The vase-shaped molecule looks like a ladder that is bent in the middle and it contains three major parts: stem, turn and loop regions (Fig. 1c, d). SL1067 maintains its structure through a variety of interactions, which include base pairing, base–base stacking, and base-2Nap stacking. Throughout the paper, the entire 2Napthyl-modified dU nucleotide is referred to as 2Nap-dUX, the 2Napthyl moiety as 2NapX, and the uridine base as dUX, where X is the nucleotide number within SL1067.

**Motifs and structural elements supporting SL1067 structure.**
*Stem region.* Despite the relatively small size of SL1067, the 2Nap modifications enable it to form an elaborate, compact, and well-defined structure. Within the stem of SL1067, nucleotides at the 5′ and 3′ termini make two Watson–Crick pairings (dC1-dG22, dG2-dC21), representing the only cannonical duplex region in the entire molecule. Adjacent to this tiny duplex there is an unusual two-step zipper motif that is formed by 2Nap-dU3 and 2Nap-dU20 (Fig. 2a, *circled* with a *green oval*). These modified nucleotides exhibit reciprocal symmetry, such that the 2Nap from one nucleotide forms a π-stack with the uridine from the other, thereby resembling the teeth of a zipper. This zipper motif effectively unwinds the surrounding helix, resulting in the complete absence of helical character within SL1067, even though 2Nap3 and 2Nap20 behave as pseudo bases. We have observed a similar zipper motif previously with benzyl-dU modifications (also in the context of an unwound helix)[21, 25], so this appears to be a recurring motif with modified aptamers containing hydrophobic, aromatic side chains oriented in a manner that allows the formation of pseudo-base-pairs.

*Turn region.* The turn region of SL1067 contains three distinct motifs: a U–A=U base triple, a G-quadruplex and an A=A base pair (Fig. 2). The dU3–dA16–dU8 base triple involves a Watson–Crick pair between dU8 and dA16 and a *trans* Watson–Crick/Hoogsteen[26] pairing between dU3 and dA16 (Fig. 2b). The dG4–dG19–dG17–dG6 quadruplex is unprecedented, in that it adopts an "N" shape that is composed of diverse non-Watson–Crick base pairings[27, 28]. For example, dG19 and dG4 form a *trans* Hoogsteen/Watson–Crick base pair, while dG17 and dG6 form a type of *trans* Watson–Crick/Watson–Crick base pair (Fig. 2c). At the center of this array is the dG4–dG17 pair, which contains a *trans* sugar-edge/sugar-edge base pair (Fig. 2c). Finally, at the bottom of the SL1067 turn, there is a dA5–dA18 base pair that is formed by *trans* Hoogsteen/Hoogsteen pairing (Fig. 2d, Supplementary Table 1).

*Diverse strategies for maintaining the structure.* In addition to the motifs described above, SL1067 employs other strategies to establish its stability and structural integrity, most of which involve stacking and clustering of hydrophobic groups, particularly involving the 2Nap moiety. The deoxyribose of dG2 participates in sugar-π stacking with 2Nap15, which mimics the carbohydrate–π interaction between sugars and aromatic amino acids[29] (Fig. 3a). This stacking interaction appears to function as a snap that connects the stem and loop portions of SL1067, enabling it to maintain a vase-like shape.

Nucleotide dA9 forms a base pair with dU15 while also engaging in a stacking network in which the 2Nap-dU10 base is sandwiched between 2Nap7 and dA9 (Fig. 3b, e). This network is strengthened by an additional interaction between the O4′ of dU8 and the dA9 base (Fig. 3b). A similar sugar-mediated arrangement is observed in the dU14/2Nap10 stacking interaction, which

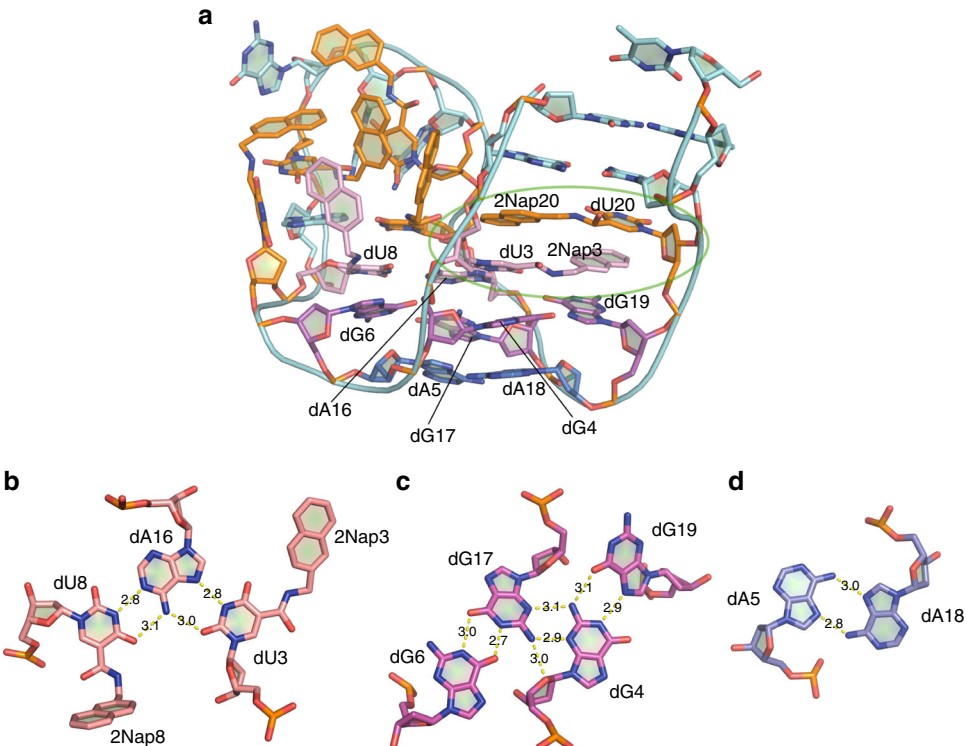

**Fig. 2** Important structural motifs in SL1067. **a** Overall structure of SL1067 with the nucleotides that participate in the formation of three important motifs colored as mentioned below. The zipper motif is indicated by a *green oval*. **b** The dU3-dA16=dU8 triple is colored *pink*. **c** The G-quadruplex is colored *magenta*. **d** The bottom dA5–dA18 pair is colored *blue*

is capped by a contact with the O4′ group of dG13 (Fig. 3c). Finally, the dU7 and 2Nap8 residues form a classical edge-to-face π interaction (Fig. 3d).

There are numerous multilayer base–base stacking interactions within the SL1067 structure such as the dU15-dA9 base pair which forms a four-layer stack with dA16-dU8, dG17-dG6 and with dA5 (Fig. 3e). Interestingly, two additional sugar-capping interactions strengthen these multilayer stacks. The O4′ of dU15 caps an interaction with dA16, and O4′ of dA5 caps the dG6 base (Fig. 3e). Similar interactions are observed between dC1-O4′ and the dG2-base, and dC21-O4′ and the G22-base (Supplementary Fig. 2). In these capping interactions, the distance between sugar atoms and their base partners ranges from 2.7 to 3.1 Å, likely providing additional stabilization for maintenance of SL1067 structure.

*Loop region.* The loop region of SL1067 is comprised of 2Nap-dU10, dG11, dG12, dG13, and 2Nap-dU14 nucleotides. The dG12 and dG13 interact with each other through a π–π stacking interaction between the base rings (Fig. 3f). Instead of forming a triple stack with dG12 and dG13, dG11 points out of the SL1067 backbone to interact with IL-1α, as described later.

*A hydrophobic cluster on the surface.* A particularly prominent structural element within SL1067 is the large hydrophobic cluster along its surface. This cluster is composed of the 2Nap groups from dU7, dU8, dU10, dU14, and dU15 (all modified nucleotides except the two forming the zipper motif) and an additional dG13 base. Interactions among these nucleotides include: (1) hydrophobic contact between 2Nap14 edge and dG11 base; (2) edge-to-face π interaction between 2Nap10 and 2Nap15; (3) stacking between 2Nap8 and the linker of 2Nap-dU10; (4) edge-to-edge stacking between 2Nap7 and 2Nap10; and (5) stacking between 2Nap10 and dU14 base and linker (Fig. 3d). This cluster comprises a large hydrophobic surface that interacts directly with IL-1α, as discussed below.

*SL1067 is a compressed nucleic acid.* The surface-area-to-volume ratio of available RNA and DNA structures was computed, and was found to range between 0.490 and 0.557 Å$^{-1}$ (Supplementary Table 2). This analysis shows that aptamers, ribozymes, ribosomes and many other molecules have a degree of compaction that falls within a relatively narrow range. It was therefore particularly striking when we determined that the surface-area-to-volume ratio for SL1067 is 0.448 Å$^{-1}$. This value is markedly smaller than the most compact nucleic acid or modified nucleic acid in the database (modified DNA aptamer for NGF, 0.490 Å$^{-1}$[25]) and it is 21% smaller than the mean value for all calculated nucleic acids (0.544 Å$^{-1}$) (Supplementary Fig. 3). These data indicate that SL1067 is a highly compressed structure, and they suggest that the 2Nap substituents aid in extreme compaction.

*Unusual sugar pucker and torsion angles.* Among the SL1067 nucleotides, 15 of the 23 deoxyribose sugar rings adopt C2′-endo-like sugar puckers, while the others adopt a C3′-endo-like conformation (Supplementary Table 5). This observation is consistent with the conformational preferences observed in other DNA aptamers that are bound to target proteins[21]. The neighboring phosphate distances vary more widely (5.2–7.7 Å) than normal DNAs and RNAs (5.8–7.0 Å)[30], providing the necessary flexibility for the formation of unusual structural motifs and explaining the lack of helical character (Supplementary Table 3).

Many of the nucleotides, both natural and modified, have unusual glycosidic torsion angles ($\chi$), and they adopt atypical sugar-phosphate backbone conformations that are most readily described by the pseudo-torsion angles $\eta$ and $\theta$[31] (Supplementary Table 3). An $\eta$–$\theta$ plot[32] shows that 80% of SL1067 nucleotides fall outside the range of the standard helical geometries[21] (150° < $\eta$ < 190°, 170° < $\theta$ < 240°), which is consistent with the heavily underwound structure (Supplementary Fig. 4). For instance,

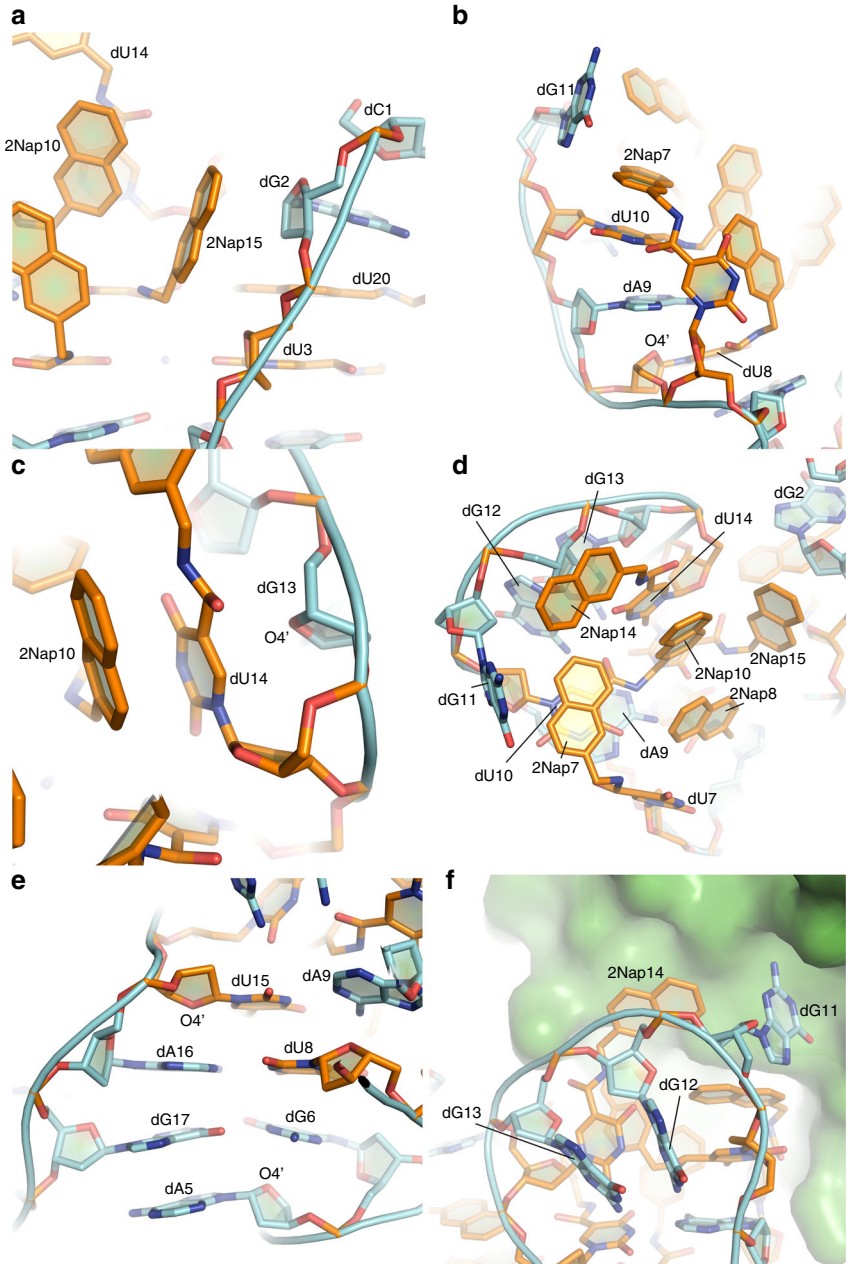

**Fig. 3** Key elements of the SL1067 structure. **a** The snap formed by 2Nap15 and the sugar ring of dG2 combines the stem and loop regions of SL1067. **b** Multilayer stacking of 2Nap7, the base of dU10 and the base of dA9 is capped with O4′ of dU8. **c** O4′(dG13) caps the Nap-base stack between dU10 and dU14. **d** The hydrophobic cluster on SL1067. The dU7 and 2Nap8 residues form a classical edge-to-face $\pi$ interaction. Two other pairs of edge-to-face $\pi$ interactions are formed between 2Nap10 and 2Nap15, and 2Nap14 and dG11. 2Nap8 makes face-to-face stacking with 2Nap10. **e** Four-layer base stacking is capped with the O4′ of the dU15 sugar ring on the top and the O4′ of dA5 sugar ring on the bottom. The stacking is further strengthened by the base pairings of dA9-dU15, dA16-dU8 and dG17-dG6. **f** In the loop region, the dG12 and dG13 base rings form face-to-face stacking and dG11 protrudes toward a hydrophobic pocket on the IL-1α surface

dG11 has $\eta$ and $\theta$ values of 77.0° and 308.3°, which are typically found in stacked turn regions of RNA[31, 33], and these play an important role in making the turn of loop region in SL1067 (Fig. 3d). Similarly, 2NapdU7 ($\eta = 36.8°$ and $\theta = 215.1°$) is engaged in the formation of an unusual bulge that not only facilitates a *syn* conformation that would otherwise cause steric interference, but also presents the nucleobase in an orientation that is perpendicular to the surrounding bases (dA9 and dU10) (Fig. 3b). This conformation extends the 2Nap group into the hydrophobic cluster on the SL1067 surface, forming a stacking network with dU10 and dA9 (Fig. 3b). The unusual conformation

of the dG17 nucleotide ($\eta = 160.3°$ and $\theta = 145.0°$), together with the *syn* orientations of dA16 and dG6, is essential for making the unconventional G-quadruplex observed in this structure (Figs. 3e, 2c). Furthermore, dA5 ($\eta = -150.7°$ and $\theta = 92.5°$) forms an A–A pair with dA18 (Fig. 2d) and both nucleotides have severely twisted glycosidic bonds ($\chi$ angles of −177.3° and 162.2°, respectively), making the sugar rings almost parallel to the purine rings. Similar behavior is observed in dU8, dU14, and dU15, which participate in the hydrophobic cluster. Several additional nucleotides adopt the *syn* conformation to facilitate base pairing (Supplementary Table 4).

The uncommon configurations of bases, sugars and modified bases within the SL1067 structure underscore the numerous ways that modified aptamers can participate in hydrophobic interactions by exploiting various conformations of the sugar-phosphate backbone. Together, the results add rich new structural diversity to the functional repertoire of modified aptamers, and that of nucleic acids in general.

**IL-1α and the SL1067 binding interface**. The interface between SL1067 and its target protein IL-1α involves an extensive network of hydrophobic interactions and, in contrast to other modified aptamer complexes, a large number of polar contacts (Fig. 1b, Supplementary Table 4). It is likely that much of the binding energy between SL1067 and IL-1α is conferred by interactions between a cluster of hydrophobic moieties on the surface of SL1067 (formed by five 2Nap modified nucleotides) and hydrophobic regions of amino acids on the IL-1α surface (Fig. 4a). Methylene groups of Met15 form $\pi$ interactions with the faces of 2Nap10 and 2Nap15, while methylene groups of the Arg16 side chain interact with the edges of 2Nap10 and 2Nap15 (Fig. 4c). Two sets of intermolecular contacts, involving Ile18 interaction with the face of 2Nap14, and the Ile68 interaction with the face of 2Nap7 have a remarkably similar geometry (Fig. 4b). Additionally, there is an edge-to-face $\pi$ interaction between the Trp113 side chain and the face of 2Nap14 (Fig. 4b), and interactions between the methylene groups of Lys60 and the face of 2Nap8 (Fig. 4c). Another characteristic of the IL-1α/SL1067 recognition interface is the protrusion of nucleobase dG11 from the SL1067 loop region. This conformation enables dG11 to insert within a small hydrophobic pocket formed by Lys67, Ile68, and Trp113, making a cation–$\pi$ interaction with the aromatic side chain of Trp113 (Fig. 4a). Although similar $\pi$ bonding/stacking interactions have been seen in previous aptamer/protein complexes[34–36], adjacent nucleobases are not significantly involved in reinforcing the local architecture. However in the case of SL1067, the adjacent 2Nap-dU units play a special role. The 2Nap7 and 2Nap14 groups are in close proximity to dG11, where they appear to stabilize the dG11 binding pocket by making non-polar contacts with the amino acids of the pocket (Ile68 and Trp113) and resulting in a tight interaction cluster near the pocket.

In conjunction with the numerous hydrophobic contacts, several polar interactions are also observed at the protein–SL1067 interface, and these may contribute to binding affinity and specificity. The polar interaction network includes highly ordered water molecules, as reported for other modified aptamer complexes[23]. A striking feature of the overall complex is that SL1067 can be visualized as a hand clutching a baseball, in which the palm mediates hydrophobic contacts and four fingers mediate polar interactions.

An unusual attribute of the IL-1α/SL1067 interface is a complicated hydrogen bonding network that contains well-ordered water molecules surrounding a monovalent cation (Fig. 5a and b). Metal ion-oxygen distances range from 2.3 to 2.6 Å and the ligands are arranged in an octahedral geometry, as typically observed for sodium ion (Fig. 5a, Supplementary Table 5)[37]. This assignment is consistent with the presence of sodium in the crystallization buffer. Three of the ligating water molecules form additional H-bonds with the surrounding residues, further tightening the H-bond network (Fig. 5a and b). Additionally, the Watson–Crick face of 2Nap-dU7 forms two H-bonds with Ser61, which are buttressed by another interaction between 2NapdU7 and Lys60 mediated by water molecule W4 (Fig. 6a), while the Watson–Crick face of G11 extruded from the loop (Figs. 3f, 4a) forms two hydrogen bonds with Ile68 and Trp113 (Fig. 6b). These regions of H-bonding, together with the

ion-mediated network, comprise the "fingers" of the polar interaction network, while W5 mediates hydrogen bonding between Arg16 and the carbonyl oxygen of the amide linker of dU14 (Fig. 6c), representing the thumb.

**The cluster-forming 2Nap-dUs are structurally essential**. Given the compactness of SL1067 and its well-defined binding interface, we explored the specific requirements of the 2Nap group for the IL-1α/ SL1067 interaction at each modified position. For most of the seven modified bases, high-affinity binding of SL1067 is exquisitely sensitive to the precise disposition of the naphthyl group, in terms of the location of substitution on the naphthyl ring and its distance from the base. For example, increasing the linker length by just one methylene spacer (2Ne-dU) has a uniformly negative effect on binding (Supplementary Fig. 5). Similarly, moving the linker attachment from the 2-position to the 1-position on the naphthyl ring (1Nap-dU and 1Ne-dU) is tolerated only within the zipper motif, at positions 3 and 20 (within 2-fold change). Positions 3 and 20 are also tolerant to other types of substitutions capable of maintaining the reciprocal stacking of the zipper (Nap-dU, Bn-dU, and MBn-dU). Notably, dT and the non-planar ring substitutions were not tolerated. As a whole, these results underscore the discriminating nature of the spaces occupied by the 2Nap groups in which size, orientation and chemical makeup of the modifications are all essential to forming the complementary interface between SL1067 and IL-1α.

**SL1067 is highly specific towards IL-1α but not IL-1β**. Although the mature forms of IL-1α and IL-1β share only 22% sequence identity, the two cytokines are structural homologs (RMSD 1.56 Å) and they reportedly share the same receptor[38]. Given their structural similarity, we tested the affinity of SL1067 for IL-1α and IL-1β using equilibrium binding experiments. While SL1067 binds to IL-1α with an affinity of 7.3 nM (Fig. 7a), it displays no detectable binding to IL-1β at protein concentrations up to 100 nM thereby indicating that SL1067 is highly specific for IL-1α. The structural superposition of IL-1β (PDB: 1ITB, IL-1β/IL-1 receptor complex) with IL-1α from the IL-1α/ SL1067 complex reveals several differences that may account for the specificity of SL1067. IL-1β lacks the first β-strand that is present in IL-1α where Met15, Arg16, and Ile18 make both polar and non-polar contacts with SL1067 (Fig. 1b). Moreover, compared with the SL1067 binding region of IL-1α, IL-1β has a much smaller hydrophobic area and different surface topology (Supplementary Fig. 6a), making it an unsuitable target for SL1067.

**SL1067 competes with the IL-1 receptor for IL-1α**. Although a structure of the IL-1α/IL-1RI (IL-1α receptor) complex has yet to be solved, a superposition of the IL-1α/SL1067 complex with IL-1β/IL-1RI (PDB ID 1ITB) enables one to make several predictions, given that IL-1α and IL-1β are structurally and functionally similar. This analysis suggests that SL1067 and the IL-1RI receptor Domain III might recognize the same surface on IL-1α (Supplementary Fig. 6c). Further evidence from mutational experiments indicates that Arg16, Ile18, Asp64, Asp65, and Trp113 are engaged in its receptor binding[39], and given that these same residues overlap with the SL1067 interface, it suggested that SL1067 might be capable of competing with IL-1α receptor.

To test this, we performed competition binding experiments using the extracellular domains of recombinant IL-1RI and IL-1RII. Both IL-1RI and IL-1RII are capable of competing with SL1067, resulting in $K_i$ values of $2.1 \times 10^{-8}$ M and $2.9 \times 10^{-7}$ M, respectively, for IL-1α binding (Fig. 7b). These results are consistent with reports that IL-1α binds IL-1RI with higher

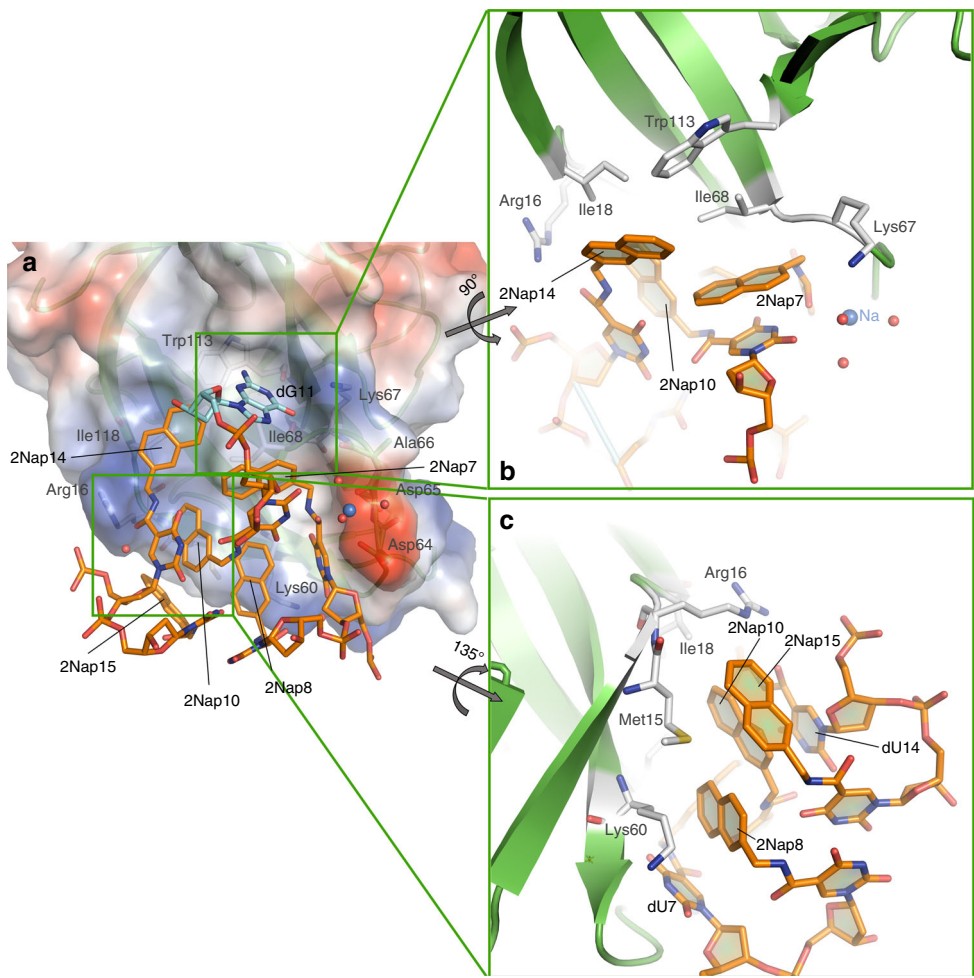

**Fig. 4** The hydrophobic interactions along the IL-1α/SL1067 interface. **a** The IL-1α surface is transparent. The SL1067 residues that are evolved in the hydrophobic contacts with IL-1α are rendered as *sticks*. **b** The Ile68 side chain and 2Nap7 form a stacking interaction similar to Ile18 and 2Nap14. **c** The Met15 side chain forms edge-to-face π interactions with 2Nap10 and 2Nap15, separately. The Arg16 side chain forms edge-to-edge contacts with 2Nap10 and 2Nap15, respectively. Modified nucleotides are labeled in *black* and amino acid residues forming the binding interface are labeled in *gray* throughout. The two *gray arrows* indicate the direction and degree of rotation for each inset view

affinity than IL-1RII[40, 41] and they demonstrate that SL1067 and the native receptor recognize the same epitope on IL-1α.

We next evaluated the ability of an SL1067 derivative containing a 5′-biotin (SL1067-b) to inhibit IL-1α-mediated secretion of IL-6 and IL-8 in cell culture[42]. Treatment of HS27 or HuVEC cells with recombinant IL-1α stimulates the IL-1RI signaling pathway and yields an IL-1α dependent increase in IL-6 and IL-8 secretion. Incubation of cells with SL1067-b prior to IL-1α stimulation results in a dose-dependent decrease in IL-6 and IL-8 secretion in both HS27 and HuVEC cells[42, 43], with IC$_{50}$ values of 23 nM and 3 nM, respectively (Fig. 8). No decrease in IL-6 or IL-8 secretion was detected using an SL1067 control sequence in which nucleotides 7–15 of the binding interface are scrambled to prevent binding to IL-1α (SL1067-scrm). These results confirm that SL1067 is a potent antagonist of IL-1RI signaling.

## Discussion

IL-1α is the major cytokine that initiates sterile inflammation during cell necrosis and tissue damage[44, 45] and it is involved in many aspects of cancer development including tumorigenesis, tumor invasiveness, and metastasis[46, 47]. The detailed, high-resolution structure of IL-1α provides a framework for explaining the role of this cytokine in tissue maintenance and

disease. For example, previous mutational studies have suggested receptor binding sites on IL-1α[39] and implicated specific amino acid side chains in receptor recognition. It is now possible to map these onto the IL-1α structure and understand the interplay among interacting residues. The structure can be used to interpret oncogenic mutations and SNPs[48], shedding light not only on interaction strategies with its own receptor, but also with the co-receptor IL-1RAcP, thereby explaining why certain mutations are signaling-deficient while retaining affinity with the receptor. For example, after IL-1α binds to the primary receptor IL-1R1, this complex recruits an accessory protein (IL-1RAcP) to initiate downstream events. In the crystal structure of the trimeric IL-1β/IL-1R1/IL-1RAcP complex (PDB ID 4DEP), the amino acid cognate of Asp151 in IL-1β does not contact the receptor IL-1R1, but rather makes several contacts with residues on IL-1RAcP[49]. Because our new structure makes it possible to visualize the side chains of IL-1α, it is now clear that Asp151 is in a position to interact only with IL-1RAcP, rather than the receptor, as observed for the cognate amino acid in IL-1β. Going forward, the high-resolution structure of IL-1α and its complex with the specific aptamer SL1067, can be used to interpret whether IL-1α or IL-1β is involved in a given biological process, potentially providing insight into the mechanism of IL-1α function. Our data show that SL1067 binds IL-1α but not

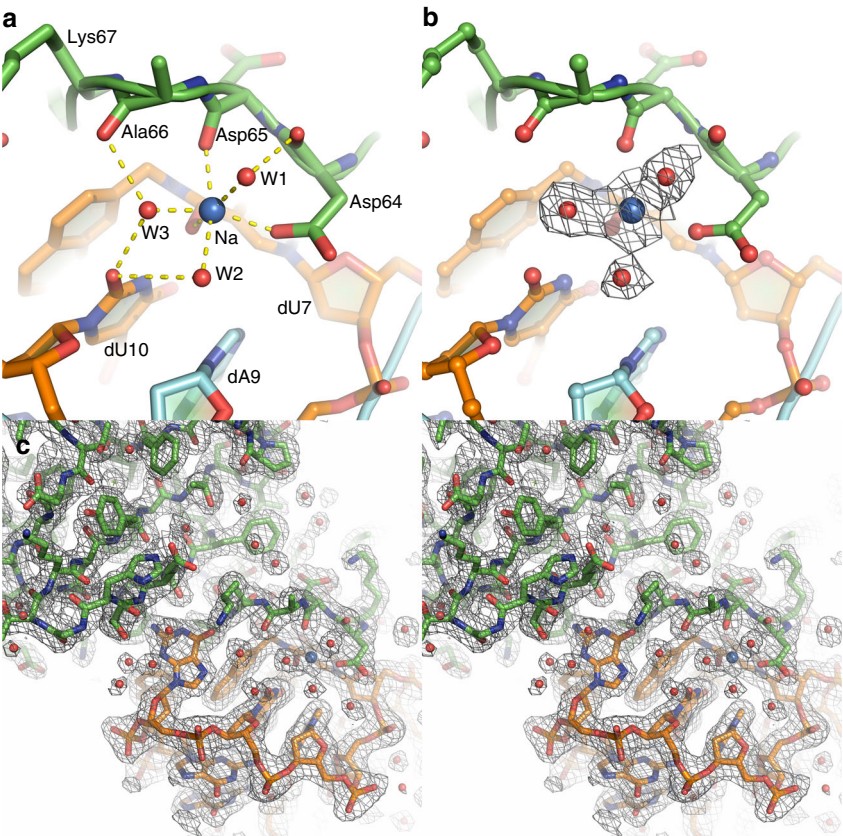

**Fig. 5** The H-bond network along the IL-1α/SL1067 binding interface is mediated by a Na$^+$ ion and three water molecules. **a** W1, W2, W3, OD1(Asp64), O (Asp65), and O21(2Nap-dU7) are coordinated by a sodium ion. Additional H-bonds are formed with the surrounding residues from either IL-1α or SL1067: W1-O (Asp64), W2-O2 (2Nap-dU10), O(Ala66)-W3-O2(dU10). The lengths of these H-bonds are listed in Supplementary Tables 4 and 5. **b** The same view of the H-bond network as shown in **a**. 2Fo–Fc electron density for the Na$^+$ ion and surrounding water molecules is shown in gray mesh and contoured at 1σ. **c** Wall-eyed stereo image of the IL-1α/SL1067 crystal structure. Electron density from the final 2Fo–Fc map contoured at 1.0σ. SL1067 is colored *orange* and the protein is colored in *green*. *Red spheres* depict associated water molecules

IL-1β, and so it could be a useful tool for mapping out interaction networks that are specific to the IL-1α signaling pathway[50].

The remarkable compaction of SL1067 and its unusual capabilities are attributable to the incorporation of 2Nap moieties throughout the structure. SL1067 is the only structurally and biochemically characterized aptamer that contains 2Nap modifications, which were incorporated at all deoxyuridine residues. This is significant because the 2Nap moiety is a completely synthetic side chain, having no cognate among natural amino acids. It therefore represents an artificial hybrid biomolecule that unites the valuable attributes of nucleic acids with the functional diversity of synthetic organic molecules. The unusually hydrophobic 2Nap groups allows folding into an exceptionally complex structure despite the short length of SL1067, which is the shortest modified aptamer that has been structurally characterized or shown to be functional[23–25]. The presence of the 2Nap moiety may explain why the SELEX against IL-1α was successful at all, given that IL-1α is an exceptionally difficult SELEX target. From ten separate SELEX experiments with seven diverse modified nucleotides, including single and double ring aromatics as well as fluorobenzyl and tyrosyl groups, only one SELEX experiment yielded a high-affinity ligand. While it is not obvious from a study of the structure and surface of the protein what makes IL-1α a difficult target for SELEX, we can speculate that it may be due to a combination of factors including the small, compact nature of the protein, a low isoelectric point of 5.3 and the requirement for long-lived complexes imposed by the kinetic challenge step in our selection protocol. The relatively higher hydrophobic character of

the naphthyl group compared with the other modified nucleotides we tested[51], coupled with its aromatic nature, may render this side chain especially effective for SELEX.

Not all attributes of SL1067 are completely unprecedented, as it shares several similarities with the (PDGF)-B, IL-6 and NGF-modified aptamers[23–25], which are also short DNA chains (24–32 nt) containing modified nucleotides (7–10) that fold into complex structures that avidly bind specific protein targets. In all of these cases, the modified side chains cluster together to stabilize a compact local conformation.

Features that are unique to SL1067 are due, in part, to inclusion of the 2Nap moiety and its connection by a highly flexible methyl-carboxamide linker. The large size of the 2Nap group enables SL1067 to make hydrophobic contacts with several residues simultaneously, resulting in a variety of interactions, including 2Nap-sugar stacking, edge-to-face π stacking, 2Nap-base stacking and even multilayer stacking between 2Nap and nucleobase surfaces. In SL1067, these interactions join the stem and loop regions together, creating an extensive hydrophobic cluster on the surface that provides a platform for binding with IL-1α.

In addition to these hydrophobic effects, the conformations of SL1067 nucleotides are highly unusual. Aberrant backbone torsion angles (η and θ) and base glycosidic torsion angles (χ), generate local structural features that resemble motifs within complex RNA structures[52]. SL1067 exhibits a greater percentage of *syn* conformers (41%) than any previous aptamer structure[21]. These *syn* conformations enable SL1067 to form unusual base

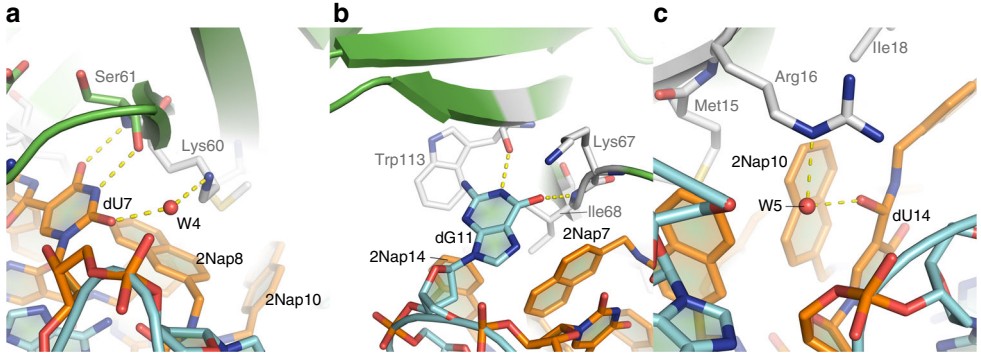

**Fig. 6** Polar interactions at the binding interface. **a** The Watson–Crick face of dU7 participates in two H-bonds with Ser61. An additional H-bond with Lys60 is mediated by W4. **b** The Watson–Crick face of dG11 forms H-bonds with Ile68 and Trp113. **c** W5 mediates hydrogen bonding between Arg16 and 2Nap-dU14

pairs, and give rise to the unusual G-quartet that is observed within the structure. Several of the *syn* nucleotides in SL1067 engage in lone pair–π interactions (lp–π) forming five "Z-like" steps[53] (dC1-dG2, dA5-dG6, 2NapdU8-dA9, 2NapdU15-dA16, dC21-dG22). In lp–π interactions, the ring oxygen of the 5′-sugar interacts with the aromatic ring of the 3′-nucleobase[54]. Interestingly, SL1067 contains an example of an ApG Z-like step in a DNA structure[53]. Furthermore, the five Z-like steps account for 23% of dinucleotide steps in SL1067, which far exceeds the ≈0.1% frequency of Z-like steps found for all dinucleotide steps in the PDB[53]. Whether the lp–π interaction is a significant energetic driver for folding is not clear[55], but in SL1067 these steps are proximal to regions of extensive base pairing and long-range interaction.

The main chain carbon atoms of IL-1α retain the same position in the presence and absence of SL1067, indicating that the aptamer does not distort the conformation of the protein, and that the aptamer adapts to accommodate the native structure of IL-1α. The interaction interface involves a combination of polar and hydrophobic interactions that form along the surface of IL-1α. The sum of H-bonds and charge–charge interactions (direct bonds only) is 0.91 per 100 Å$^2$ for SL1067, indicating a significantly greater role for polar interactions than observed for other modified aptamer–protein complexes (0.57–0.82 per 100 Å$^2$). For example, when compared with the water-mediated H-bonds in the NGF-modified aptamer structure[25], there are far more water molecules at the IL-1α/SL1067 interface, three of which take part in the unique sodium ion coordinated H-bond network. The quality of the electron density and the relatively low B-factors along the interface show that the H-bond network is highly ordered and serves as a major architectural component for specific binding of IL-1α and SL1067. The observed hydrophobic interactions at the IL-1α/SL1067 interface are mainly edge-to-face π interactions, exemplified by the base of dG1, which protrudes into a small hydrophobic pocket on the IL-1α surface.

Overall, SL1067 has a smaller interaction surface area (657 Å$^2$) compared with the other three modified aptamer complexes that have been structurally characterized (1097–1248 Å$^2$). However, its ligand efficiency of 0.21 kcal mol$^{-1}$ per non-H contact atom is significantly larger than other modified aptamers (0.12–0.18 kcal mol$^{-1}$ per non-H contact atom)[21, 23]. Accordingly, the calculated value of the shape complementarity (Sc) index[56, 57] for IL-1α/SL1067 is 0.725, which is at the lower end, but within the range reported for the three other modified aptamers (0.72–0.80)[21]. Together these measures imply the somewhat reduced binding affinity of SL1067 compared to previously reported modified aptamers may simply result from the reduced number of hydrophobic contact atoms (54 for SL1067 compared to 75, 78, and 131 for other published modified aptamers[23–25]). They also

provide evidence that, in spite of the substantial hydrophobicity of the 2Nap moiety, this modification significantly contributes to the binding energy and is capable of forming a highly complementary interface with the target protein.

Amino acids that participate in the IL-1α/SL1067 interface overlap with those that are required for receptor binding, indicating that SL1067 and the IL-1RI Domain III share a common epitope on IL-1α (known as site B (Supplementary Fig. 5)). Indeed, both IL-1RI and IL-1RII act as competitive inhibitors for the binding of SL1067 to IL-1α (Fig. 7). These findings suggest that SL1067 should act as an antagonist for IL-1α signaling, which was confirmed in cell-based assays (Fig. 8). Not only does SL1067 mimic and compete for binding by the native receptor, the affinity of SL1067 is significantly higher than IL-1RI and IL-1RII, based on competition assay (Fig. 7), demonstrating that a synthetic ligand can be selected for potency superior to that achieved by the natural ligand.

It is notable that binding "site B" is present on both IL-1α and IL-1β, but not on IL-1Ra (IL-1R antagonist)[58]. Thus, SL1067 would not be expected to inhibit the activity of IL-1Ra and the two ligands may act synergistically to prevent IL-1RI signal activation. Although the "site B" epitope on the IL-1α and IL-1β structural homologs overlap, they differ in local surface architecture and hydrophobicity for the two cytokines. These differences explain why SL1067 binds specifically to IL-1α, and they suggest that IL-1RI may use different strategies to recognize IL-1α and IL-1β. An alignment of the IL-1α/SL1067 structure and the NMR structure of an IL-1α/S100A13 complex shows that the SL1067 and S100A13 bind on opposite faces of IL-1α, underscoring the fact that the small IL-1α protein has multiple epitopes for molecular recognition and suggesting that SL1067 would not affect the formation of an IL-1α/S100A13 complex, which is involved in the release of IL-1α through a non-classical pathway[20].

SL1067 is a potent tool for studying various processes in which IL-1α is engaged but the specifics remain undefined, e.g., Caspase-1 dependent IL-1α secretion[59], the potential for IL-1α/IL-1RI complex to function as an autocrine growth factor[38], and the regulation of functional IL-1α activation by IL-1RII in necrosis[17], as well as its role in numerous pathogenic pathways.

The scope of human diseases for which IL-1α involvement has been implicated is extensive and diverse and includes Type 2 diabetes[18], cancer[12, 13, 60], cancer cachexia[61], leukemia[62], psoriasis[63], vascular disease[15, 17], and scarring acne vulgaris[18]. Therapeutic intervention of IL-1RI signaling via antagonism of IL-1α is therefore of great interest.

In this study we show that the modified DNA aptamer SL1067 displays high affinity and selectivity for its IL-1α target, and that it behaves as an inhibitor that can block downstream signaling

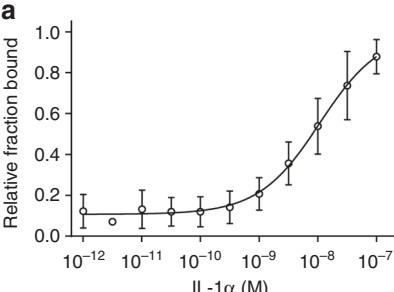
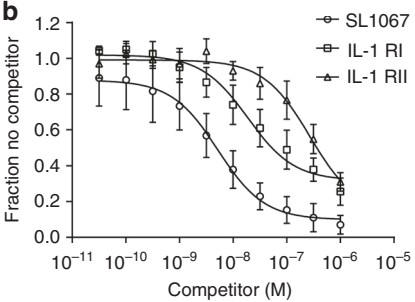

**Fig. 7** Determination of the binding affinity of SL1067 to IL-1α and the competition of IL-1α receptors with SL1067. **a** Shown is a four-parameter global fit (mean ± S.D., $n = 4$) of the relative fraction bound vs. IL-1α concentration. The mean (±S.D.) binding dissociation constant ($K_d$) determined from the four independent experiments is $7.3 ± 1.5$ nM. **b** Displacement of radiolabeled SL1067 by unlabeled competitors SL1067 (*open circles*), IL-1 RI (*open squares*) or IL-1 RII (*open triangles*). Shown is a global fit (mean ± S.D., $n = 3$) to a one-site competition model of competitor concentration vs. fraction no competitor. The mean (±S.D.) inhibitory constants ($K_i$) from three determinations are: SL1067, $K_i = 4.3 ± 1.5$ nM; IL-1 RI $K_i = 21 ± 9.8$ nM and IL-1 RII $K_i = 290 ± 130$ nM

pathways. These findings suggest that SL1067 is a valuable tool for investigating signaling cascades and inflammatory responses. In addition, our results underscore the potential clinical value of SL1067 for a variety of disease indications. The ease of synthesis, stability, small size and high specificity of SL1067 provide a valuable foundation for the development of new therapeutics that target IL-1α and related cytokines.

## Methods

**SELEX and modified aptamer synthesis.** Modified aptamers with affinity for recombinant human IL-1α protein (ProSpec) were discovered via SELEX with a slow off-rate enrichment step to select for modified aptamers with slow off-rates[64]. The DNA library included a 40-nucleotide random region in which 2Nap-dU was substituted for dT as well as a 16-nucleotide forward primer (5′-GCCACACCCT GCCCTC) and an 18 nucleotide reverse primer (5′-GAGGACACAGACAGAC AC). The IL-1α protein was partitioned on Talon Dynabeads (ThermoFisher). To obtain SOMAmers with slow off-rates, a kinetic challenge was introduced wherein protein–DNA complexes were incubated in the presence of 10 mM dextran sulfate at 37 °C with increased incubation times (15 min up to 30 min) and decreased protein concentrations in successive rounds of SELEX. Kinetic challenge was introduced in round 4 and continued through the final, seventh round, of SELEX. Selection was performed in SB18T buffer (40 mM HEPES pH 7.5, 102 mM NaCl, 5 mM KCl, 5 mM MgCl₂, 0.01% Tween-20). Modified aptamers were produced by conventional solid-phase oligonucleotide synthesis using the phosphoramidite method[65]. The modified deoxyuridine-5-carboxamide phosphoramidite reagent used for solid-phase synthesis was prepared by: condensation of 5′-O-(4,4′-dime-thoxytrityl)-5-trifluoroethoxycarbonyl-2′-deoxyuridine[66] with the appropriate [2-naphthylmethylamine] primary amine (RNH₂, 1.2 equiv.; Et₃N, 3 equiv.; acetonitrile; 60 °C; 4 h); 3′-O-phophitylation with 2-cyanoethyl-*N,N,N′,N′*-tetraisopropylphosphoramidite (1.2 equiv.; iPr₂EtN, 3 equiv.; CH₂Cl₂; −10 to 0 °C; 4 h); and purification by flash chromatography on neutral silica gel[67]. Modified aptamers were synthesized at the 1 μmol scale on a ABI 3900 DNA synthesizer with some adjustments to the protocol to account for the unique base modifications described herein. Detritylation was accomplished with 10% dichloroacetic acid in toluene for 45 s; coupling was achieved with 0.1 M phosphoramidites in 1:1 acet-onitrile:dichloromethane activated by 5-benzylmercaptotetrazole and allowed to react three times for 5 min; capping and oxidation were performed according to instrument vendor recommendations. Deprotection was effected with gaseous ammonia or methylamine under optimized pressure, time, and temperature in a Parr stainless steel reactor. Products were eluted with dI water into 2 ml siliconized screw-cap tubes, purified by preparative ion-pairing reversed phase liquid chromatography, evaporated to dryness in a Genevac HT-12 system, and desalted on GE Hi-Trap Sephadex G-25 columns. Purified products were characterized by ultra performance liquid chromatography (UPLC, Waters Acquity system), mass spectrometry (Agilent 1100 with Bruker Ion-trap detector), UV spectrophotometry, and protein binding affinity in buffered aqueous solution.

**Determination of equilibrium binding constants.** For determination of equilibrium binding constants, purified IL-1α protein was incubated at room temperature for 30 min with a 10-fold molar excess of EZ-Link NHS-Peg₄-Biotin (ThermoFisher). Free biotin was removed using a YM-3 spin column (Millipore) and the resulting protein concentration was determined using a Micro BCA assay (ThermoFisher). Equilibrium binding constants ($K_d$ values) of modified aptamers were measured in SB18T buffer. Modified aptamers were 5′ end labeled using T4 polynucleotide kinase (New England Biolabs) and γ-[³²P]ATP (Perkin-Elmer). Radiolabeled aptamers (~ 20,000 CPM, 0.03 nM) were mixed with Il-1α protein at

concentrations ranging from $10^{-7}$ to $10^{-12}$ M and incubated at 37 °C for 40 min. Bound complexes were partitioned using Dynabeads MyOne streptavidin C1 (Life Technologies) and captured on Durapore filter plates (EMD Millipore). The fraction of bound aptamer was quantified with a phosphorimager. To determine binding affinity, data were fit using the equation:

$$y = (\text{max} - \text{min})(\text{Protein})/(K_d + \text{Protein}) + \text{min}$$

and plotted using GraphPad Prism version 7.00.

**Competition binding.** Modified aptamer SL1067 was 5′ end-labeled using T4 polynucleotide kinase (New England Biolabs) and γ-[³²P]ATP (Perkin-Elmer). Competition assays were performed by incubating radiolabeled SL1067 (~30 pM) and cold competitors, SL1067, IL-1R1 (R&D Systems), IL-1RI1 (R&D Systems) at concentrations ranging from $10^{-6}$ to $10^{-11}$ M with biotinylated IL-1α (2 nM) in SB18T buffer at 37 °C for 60 min. Bound complexes were mixed with MyOne streptavidin resin, mixed briefly and captured on Durapore filter plates. The fraction of SL1067 bound was quantified with a PhosphorImager (FUJI FLA-3000). Data were quantified using Image Gauge v4.0 (Fuji Film Science Lab). Equilibrium dissociation constants ($K_i$) for the competitors were determined by nonlinear regression analysis using the equation:

$$\log \text{EC}_{50} = \log\left[10^{\log K_i} * \left(1 + \frac{\text{Radio ligand (nM)}}{\text{Hot } K_d \text{ (nM)}}\right)\right]$$

(GraphPad Prism).

**Cell assay inhibition effect of SL1067 on IL-1α signaling.** *Cell lines.* HuVEC cells purchased from Lonza were cultured in complete 2% EGM-2 media (Lonza). Positive staining of endothelial cell markers was used by the supplier to authenticate the HuVEC. Hs27 cells purchased from ATCC were cultured in DMEM media (Thermo Fisher Scientific) supplemented with 2 mM glutamine and 10% Fetal Bovine serum (Thermo Fisher Scientific). ATCC authenticated the Hs27 using an isoenzyme assay and STR analysis. HuVEC and Hs27 cells were tested for mycoplasma contamination by their respective suppliers and none was detected. All cell lines were maintained in a humidified incubator containing 5% CO₂ at 37 °C.

*Cytokine release assay.* 5000 cells per well were plated in their respective growth media in 96-well tissue culture dishes and allowed to recover for 6 h. Modified aptamer solutions were diluted in water to the concentrations ranging from $1.0 \times 10^{-6}$ M to $6.1 \times 10^{-11}$ M and triplicate cell samples were pretreated for 2 h prior to stimulation. IL-1α (PeproTech) was stored and diluted before use in accordance with the manufacturer's instructions. modified aptamer treated cells were stimulated with 0.2 ng ml⁻¹ IL-1α for 16–20 h after which supernatants were harvested and clarified by centrifugation at 200 × *g* for 5 min. The supernatants from each experiment were stored at −80 °C and subsequently assayed for cytokine levels by ELISA. Commercial ELISA kits using horseradish peroxidase linked antibodies were used according to the manufacturer's recommendations for immunological quantification of IL-6 (R&D Systems, #D6050) and IL-8 (R&D Systems, #D8000C)). HuVEC supernatants were diluted 3×–5× while Hs27 supernatants were diluted up to1000×. Optical density was read at 450 nm in a SpectraMax M5 (Molecular Devices). All values were normalized to cell number using Promega's CellTiter-Glo cell viability assay. The average value was recorded as a percent of the no modified aptamer control and was plotted as a function of modified aptamer concentration for each condition tested. To determine IC₅₀ data were fit to a four-parameter sigmoidal dose–response model (GraphPad Prism).

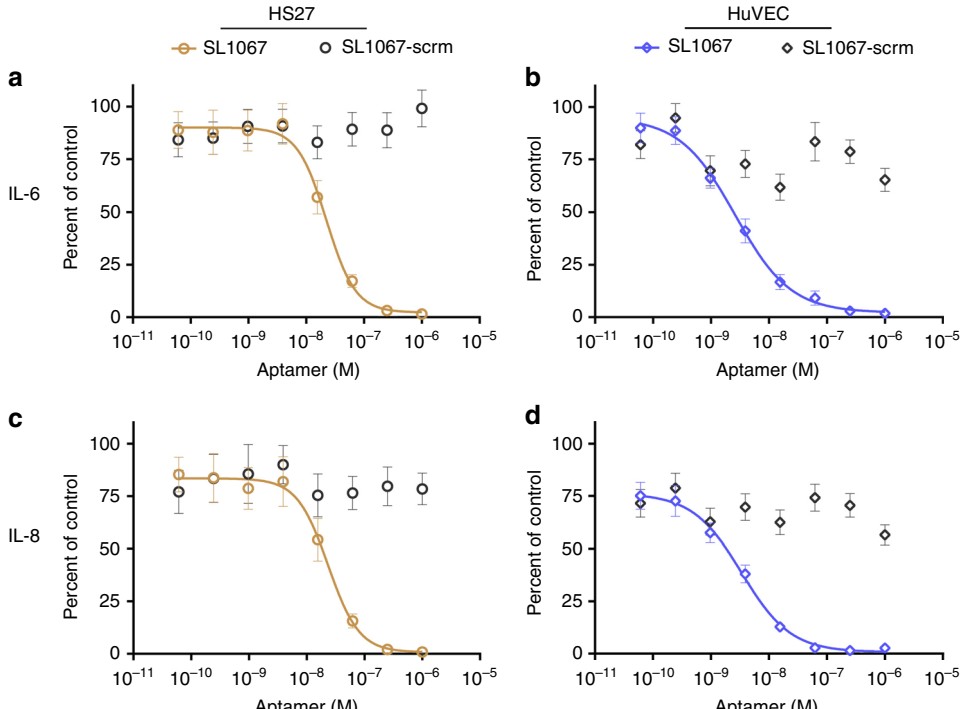

**Fig. 8** A cell-based assay shows the inhibition of IL-1α downstream signaling by SL1067 in a dose-dependent manner. Shown are the global fits (mean ± S. D., $n = 3$) to a four-parameter sigmoidal dose response model of the modified aptamer concentration vs. the cytokine secretion normalized to the percent of control (no modified aptamer condition). **a, b** SL1067 inhibits the secretion of IL-6, a downstream effector of IL-1 receptor signaling[42] in HS27 (open circles) and HuVEC (open diamonds). The mean (±S.D.) half minimal inhibitory concentration ($IC_{50}$) from three replicates are 22 ± 1.8 nM and 2.7 ± 0.73 nM in HS27 and HuVEC cells, respectively. **c, d** SL1067 inhibits the secretion of IL-8, another downstream effector of IL-1 receptor pathway[42] in HS27 (open circles) and HuVEC (open diamonds). The $IC_{50}$ (±S.D.) from three replicates are 23 ± 3.6 nM and 3.5 ± 0.70 nM in HS27 and HuVEC cells, respectively

**Cloning expression and purification of IL-1α**. Human *IL1F1* cDNA was cloned into pET-SUMO vector (ThermoFisher Scientific) using the method of TA cloning with the forward primer 5′-TCAGCACCTTTTAGCTTCCTGA and reverse primer 5′-CTACGCCTGGTTTTCCAGTATCT. The IL-1α protein was overexpressed in *Escherichia coli* Rosetta 2(DE3) cells (Millipore) through plasmid transformation of DE3 cells that were cultured at 37 °C in Luria broth (LB) medium supplemented with 25 μg chloramphenicol and 50 μg kanamycin per milliliter of LB medium. Protein expression was induced when the $OD_{600}$ reached a value of ~ 0.6–0.8 by adding isopropyl β-D-1-thiogalactopyranoside (IPTG) to a final concentration of 1 mM at which point the temperature was lowered to 16 °C. Cells were harvested 16 h later by centrifugation at 8000 × *g* for 10 min at 4 °C and stored at −20 °C. Cells were resuspended in lysis buffer (25 mM HEPES buffer pH 8.0, 300 mM NaCl, 5% (v/v) glycerol, 5 mM β-ME and 10 mM imidazole) and lysed by passing three times through a MicroFluidizer at 15,000 psi and the lysate was clarified by centrifugation at 30,000 × *g* for 30 min at 4 °C. The supernatant was applied to a nickel affinity column (Qiagen) which was equilibrated with lysis buffer. Protein was eluted with elution buffer (25 mM HEPES buffer pH 8.0, 300 mM NaCl, 5% (v/v) glycerol, 5 mM β-ME and 200 mM imidazole). The protein fraction was concentrated and diluted with lysis buffer until the concentration of imidazole was 20 mM. The protein was digested with SUMO protease for 2 h at 4 °C and then loaded onto an equilibrated nickel column. The flow-through was collected, concentrated and then purified with a HiLoad 16/60 Superdex 200 column (GE Healthcare) equilibrated with buffer containing 25 mM HEPES pH 8.0, 200 mM NaCl and 5% glycerol. Peak fractions were collected, concentrated to 20 mg ml⁻¹ and stored at −80 °C.

**Crystallization**. Solutions of SL1067 were heated to 95 °C for 10 min and then snap-cooled on ice. To make the IL-1α/SL1067 complex, IL-1α and SL1067 were mixed at a 1:1.1 ratio, in which the final protein concentration was 12 mg ml⁻¹. Crystals of the complex were grown by mixing equal volumes of precipitating solution (0.2 M NaCl, 0.1 M Na acetate, and 22% polyethylene glycol 8000). Crystals were cubic shaped and grew to full size in 2 days. After removal from the drops, crystals were flash frozen with liquid nitrogen. Diffraction data were collected at NE-CAT beamline ID-24 at the Advanced Photon Source (Argonne National Laboratory, Argonne, IL, USA). Integration, scaling and merging of the intensities were carried out with the program imosflm[68] and SCALA from the CCP4 software suite[57].

**Structure determination and refinement**. The structure of IL-1α was solved by molecular replacement with the available IL-1α PDB coordinates (PDB ID 2ILA,

with only alpha-C atoms) for the search model using Phaser in the CCP4 software suite[57]. The model was built with Autobuild in the Phenix suite[69]. The SL1067 section of the model was fit and built manually using the Rcrane[31] plugin within Coot[70]. Refinement cycles were performed using Refine in Phenix, alternating with iterative manual rebuilding in Coot[70]. A summary of data collection and structure refinement statistics is given in Table 1.

**Data availabilty**. Coordinates and structure factors have been deposited in the Protein Data Bank with accession code 5UC6. Other data supporting the findings of the manuscript are available from the corresponding author upon reasonable request.

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

## Acknowledgements

X.R. is supported by Howard Hughes Medical Institute. A.M.P. is a Howard Hughes Medical Institute Investigator. This work is supported by Howard Hughes Medical Institute. This work is based upon research conducted at the Northeastern Collaborative Access Team beamlines, which are funded by the National Institute of General Medical Sciences from the National Institutes of Health (P41 GM103403). The Pilatus 6M detector on 24-ID-C beam line is funded by a NIH-ORIP HEI grant (S10 RR029205). This research used resources of the Advanced Photon Source, a U.S. Department of Energy (DOE) Office of Science User Facility operated for the DOE Office of Science by Argonne National Laboratory under Contract No. DE-AC02-06CH11357.

## Author contributions

A.M.P. and N.J. designed research; X.R., A.D.G. and I.v.C. performed research; A.M.P., N.J., X.R., A.D.G. and I.v.C. analyzed data; A.M.P., N.J., X.R. and A.D.G. wrote the paper.

## Additional information

**Competing interests:** A.D.G., I.v.C. and N.J. are employees and shareholders of SomaLogic, Inc. The remaining authors declare no competing financial interests.

