## [Peer Review file · Nature Communications]

REVIEWERS' COMMENTS:

Reviewer #1 (Remarks to the Author):

This manuscript (MS) was a pleasure to read!

The authors report the selection of a very specific IL-1 α SOMAmer (SL 1067) which inhibits the initiation of the corresponding signal cascade - also in cell culture. And furthermore the authors report the high resolution structure of the SL 1067/IL-1 α complex.

There are a lot of remarkable results like:

- the highly specific SOMAmer consisting of only 23 nucleotides containing seven naphthyl modified building blocks five of which being involved in the target interaction.
- the high resolution structure of the complex yielding not only the structure of the SOMAmer but additionally 'en passant' the detailed structure of IL-1 α so far only available at low resolution.
- a compactness of the oligonucleotide, which has never been seen before
- a novel G-quadruplex form etc.

The results are important from basic research points of view and are of high medical relevance as the investigated IL-1 α initiated signaling passway is or might be involved in the regulation of the immune system and in illnesses like cardiovascular disease, systemic sclerosis or cancer.

The convincing data are presented in a nice and clear manner; figures are excellent.

There are, however, some points which might be addressed to possibly amend the manuscript in my opinion:

- Line 37: Why do the authors not introduce here the abbreviation/term „SOMAmer“?
- Line 98: Would it not be better to replace „2Nap-dUX“ by „2NapX-dUX“ as the latter is the term used in the figures etc.
- Line 471: Sentences like this one „The recombinant plasmid was transformed into E. coli Rosetta ...“ one can read in plenty of manuscripts or papers - even in excellent ones like the one discussed here. However, the sentence contains an incorrectness: One can transform i.e. current from 220 volts to 24 volts or vice versa, but one cannot transform a plasmid into an E. coli cell. Rather the E. coli cell is being transformed by taking up the plasmid!
- Line 473: The authors write „... Lucia broth medium supplemented with 25 μ g/ml chloramphenicol and 50 μ g/ml kanamycin“. I am sure they meant "... 25 μ g chloramphenicol and 50 μ g kanamycin per ml LB medium ..." - and not something „... per ml chloramphenicol ..." etc.

In conclusion: Excellent MS with a lot of amazing results which should not all be ruminated here.

Signed: Ulrich Hahn

Reviewer #2 (Remarks to the Author):

This work describes the crystal structure of a modified DNA aptamer (designated SL1067) in complex with the small protein IL-1 α - an essential cytokine involved in the regulation of the

immune response, and implicated in very diverse pathological conditions. This manuscript clearly deserves to be considered for publication in Nature Communications, for several reasons:

1. Owing to the synthetic hydrophobic moieties (i.e. 2-naphtylmethyl) carried by the aptamer, its architecture exhibits many unexpected and outstanding features: (i) it is the most compact nucleic acid structure ever determined, (ii) the 3D organization is remarkably complex, in spite of its small size (iii) it contains surprisingly many novel conformational features, including unusual sugar puckers and glycosidic torsions, plus a novel type of G-quadruplex.

The importance of grafted naphtyl groups for enhancing 3D complexity, to such a remarkable extent, represents an important conceptual advance in structural and synthetic biology, as well as nucleic acids chemistry.

2. The structure of the complex unveils the molecular basis for affinity and specificity of the aptamer for the cytokine. Notably, IL-1 α uses the same interface to interact both with the aptamer and with its natural cognate receptor. Cell based assays further confirmed that the aptamer acts as an antagonist for IL-1 α signaling.

3. The high affinity and selectivity of SL1067 for IL-1 α , enables effective interference of the aptamer in the downstream pathway of the cytokine. This creates strong prerequisites for using the SL1067 as a tool for investigating signaling cascades and inflammatory responses connected to the cytokine. Very importantly, the aptamer is small, stable and easy to synthesize, which makes it an excellent candidate for development of therapeutical approaches.

4. Last but not least, although the backbone structure of IL-1 α was described before, this is the first time when the structure of this cytokine is resolved to sufficient resolution to build the side chains.

Minor comments, typos:

- Line 300: "The structure .. helps explain why certain mutations are signaling-deficient while retaining affinity with the receptor." How does the structure help explain this? Perhaps one can be more explicit.

- Line 302: ".. the high-resolution structure .. can be used to test mechanistic models for cytokine function". The formulation is a bit vague and a clearer explanation might be beneficial (i.e. in which way can the structure be used to test..).

- Lines 544-545: "The surface is partially transparent." Why not just simply "The surface is transparent"?

- Line 557, related to figure 5: what sort of electron density is shown around the atoms? Is it an omit map, or Fo-Fc?

Reviewer #3 (Remarks to the Author):

In the manuscript by Ren and colleagues, the authors describe the isolation and characterization of a DNA aptamer modified with 2Nap-dUs for dT. The resulting aptamer was truncated to a 22mer and retained high affinity ($K_d=7.3nM$) for human IL-1 α (alpha). The authors determine the crystal structure of the aptamer-IL-1 α complex at 2.1Å which leads to numerous interesting observations

about the folding and structure of such modified DNA aptamers as well as potential useful structural information about IL-1a. Finally the authors provide data demonstrating that the aptamer competes with binding of IL-1 receptor to IL-1a and can inhibit IL-1a induced cytokine secretion in HS27 and HuVEC -based cell assays.

Overall the results are clear and the data is both compelling and novel. Moreover the paper is very well written and describes numerous new structural motifs for modified nucleic acid aptamers. It will be of particular interest to those studying aptamers, IL-1a and IL-1 signaling as it presents a high resolution structure of IL-1a (albeit bound to an aptamer) for the first time. The activity data is not abundant (only cell line data and no in vivo animal data) but the paper is a beautiful structural report and not a translational research article so I believe that this amount of functional activity data is more than acceptable. Therefore I would recommend the paper for publication in Nature Communications but would ask the authors to address a few minor issues that I believe would be beneficial to the reader.

1.) It would be useful for the authors to present a space filling version of the aptamer and indicate the surface on the aptamer that is in close contact with the IL-1a surface. This representation would be particularly useful in trying to understand how the 2Nap-dU aptamer is so compact yet interfaces with 657A on the surface of the protein. Moreover it would be useful if the authors would speculate on if the reduced surface interaction compared to other modified DNA aptamers they have generated results in the somewhat reduced binding affinity compared to the many sub-nM binding aptamers that they have reported previously. In particular such a model would be very useful in thinking about why the aptamer is so selective for the surface of IL-1a over IL-1b (beta) in supplemental figure 6.

2.) It would be useful for the authors to speculate on why IL-1a is such a hard target and why the 2Nap-dU modification and not other similar modifications can crack this tough target case. Is there something about the "aptagenic" epitope on the surface of the protein that is only accessible to a compact aptamer (depth, width, orientation, charge, hydrophobicity etc) or are there essential conformations that 2Nap-dU modifications allow that none of the other ten SELEX experiments with seven diverse modifications could achieve?

3.) In figure 4, it would be useful to indicate in panels b and c how the view is rotated relative to panel a. Otherwise it is difficult for the reader to follow.

4.) It would be useful to discuss how the key dG11 interactions are similar or different from unmodified or backbone modified aptamers as pi bonding/stacking interactions have been seen between such aptamers and their target proteins albeit not when the purines are presented in the context of a 2Nap-dU platform. Does this presentation allow dG11 adopt a conformation that allows it fit into a pocket that is otherwise not accessible?

REVIEWERS' COMMENTS:

Reviewer #1 (Remarks to the Author):

This manuscript (MS) was a pleasure to read!

The authors report the selection of a very specific IL-1 α SOMAmer (SL 1067) which inhibits the initiation of the corresponding signal cascade - also in cell culture. And furthermore the authors report the high resolution structure of the SL 1067/IL-1 α complex.

There are a lot of remarkable results like:

- the highly specific SOMAmer consisting of only 23 nucleotides containing seven naphthyl modified building blocks five of which being involved in the target interaction.
- the high resolution structure of the complex yielding not only the structure of the SOMAmer but additionally 'en passant' the detailed structure of IL-1 α so far only available at low resolution.
- a compactness of the oligonucleotide, which has never been seen before
- a novel G-quadruplex form etc.

The results are important from basic research points of view and are of high medical relevance as the investigated IL-1 α initiated signaling passway is or might be involved in the regulation of the immune system and in illnesses like cardiovascular disease, systemic sclerosis or cancer.

The convincing data are presented in a nice and clear manner; figures are excellent.

There are, however, some points which might be addressed to possibly amend the manuscript in my opinion:

- **Line 37: Why do the authors not introduce here the abbreviation/term „SOMAmer“?**

We made the decision not to use the term SOMAmer throughout the manuscript and instead refer to SL1067 as a modified aptamer. We feel using “modified aptamer” appeals to a broader audience.

- **Line 98: Would it not be better to replace „2Nap-dUX“ by „2NapX-dUX“ as the latter is the term used in the figures etc.**

The sentence describing nomenclature has been modified to be more consistent with the terminology used throughout the manuscript and in the figures. The sentence at line 98 now reads as follows, “Throughout the paper, the entire 2Naphthyl-modified dU nucleotide is referred to as 2Nap-dUX, the 2Naphthyl moiety as 2NapX, and the uridine base as dUX, where X is the nucleotide number within SL1067.”

- **Line 471: Sentences like this one „The recombinant plasmid was transformed into E. coli Rosetta ...“ one can read in plenty of manuscripts or papers - even in excellent ones like the one discussed here. However, the sentence contains an incorrectness: One can transform i.e. current from 220 volts to 24 volts or vice versa, but one cannot transform a plasmid into an E. coli cell. Rather the E. coli cell is being transformed by taking up the plasmid!**

This sentence has been modified on line 523 to read, “The IL-1 α protein was overexpressed in *E. coli* Rosetta 2(DE3) cells (Millipore) through plasmid transformation of DE3 cells that were cultured at 37 °C in Luria broth (LB) medium supplemented with 25 μ g chloramphenicol and 50 μ g kanamycin per milliliter of LB medium.”

• Line 473: The authors write „... Lucia broth medium supplemented with 25 μ g/ml chloramphenicol and 50 μ g/ml kanamycin“. I am sure they meant “... 25 μ g chloramphenicol and 50 μ g kanamycin per ml LB medium ...” - and not something „... per ml chloramphenicol ...“ etc.

See above

In conclusion: Excellent MS with a lot of amazing results which should not all be ruminated here.

Signed: Ulrich Hahn

Reviewer #2 (Remarks to the Author):

This work describes the crystal structure of a modified DNA aptamer (designated SL1067) in complex with the small protein IL-1 α - an essential cytokine involved in the regulation of the immune response, and implicated in very diverse pathological conditions. This manuscript clearly deserves to be considered for publication in Nature Communications, for several reasons:

1. Owing to the synthetic hydrophobic moieties (i.e. 2-naphtylmethyl) carried by the aptamer, its architecture exhibits many unexpected and outstanding features: (i) it is the most compact nucleic acid structure ever determined, (ii) the 3D organization is remarkably complex, in spite of its small size (iii) it contains surprisingly many novel conformational features, including unusual sugar puckers and glycosidic torsions, plus a novel type of G-quadruplex.

The importance of grafted naphtyl groups for enhancing 3D complexity, to such a remarkable extent, represents an important conceptual advance in structural and synthetic biology, as well as nucleic acids chemistry.

2. The structure of the complex unveils the molecular basis for affinity and specificity of the aptamer for the cytokine. Notably, IL-1 α uses the same interface to interact both with the aptamer and with its natural cognate receptor. Cell based assays further confirmed that the aptamer acts as an antagonist for IL-1 α signaling.

3. The high affinity and selectivity of SL1067 for IL-1 α , enables effective interference of the aptamer in the downstream pathway of the cytokine. This creates strong prerequisites for using the SL1067 as a tool for investigating signaling cascades and inflammatory responses connected to the cytokine. Very importantly, the aptamer is small, stable and easy to synthesize, which makes it an excellent candidate for development of therapeutical approaches.

4. Last but not least, although the backbone structure of IL-1 α was described before, this is the

first time when the structure of this cytokine is resolved to sufficient resolution to build the side chains.

Minor comments, typos:

- Line 300: "The structure .. helps explain why certain mutations are signaling-deficient while retaining affinity with the receptor." How does the structure help explain this? Perhaps one can be more explicit.

To address this, the following text has been included at line 309:

For example, after IL-1 α binds to the primary receptor IL-1R1, this complex recruits an accessory protein (IL-1RAcP) to initiate downstream events. In the crystal structure of the trimeric IL-1 β /IL-1R1/IL-1RAcP complex (PDB ID 4DEP), the amino acid cognate of Asp151 in IL-1 β does not contact the receptor IL-1R1, but rather makes several contacts with residues on IL-1RAcP⁴⁹. Because our new structure makes it possible to visualize the side chains of IL-1 α , it is now clear that Asp151 is in a position to interact only with IL-1RAcP, rather than the receptor, as observed for the cognate amino acid in IL-1 β .

- Line 302: ".. the high-resolution structure .. can be used to test mechanistic models for cytokine function". The formulation is a bit vague and a clearer explanation might be beneficial (i.e. in which way can the structure be used to test..).

To address this comment, the following text has been incorporated at line 319:

For example, our data show that SL1067 binds IL-1 α but not IL-1 β , and so it could be a useful tool for mapping out interaction networks that are specific to the IL-1 α signaling pathway⁵⁰

- Lines 544-545: "The surface is partially transparent." Why not just simply "The surface is transparent"?

The word "partially" has been deleted.

- Line 557, related to figure 5: what sort of electron density is shown around the atoms? Is it an omit map, or Fo-Fc?

Figure 5b shows the 2Fo-Fc electron density for Na⁺ ion and the water molecules surrounding it contoured at 1 σ . The figure legend has been modified at line 866 to read, "2Fo-Fc electron density for the Na⁺ ion and surrounding water molecules is shown in grey mesh and contoured at 1 σ ."

Reviewer #3 (Remarks to the Author):

In the manuscript by Ren and colleagues, the authors describe the isolation and characterization of a DNA aptamer modified with 2Nap-dUs for dT. The resulting aptamer was

truncated to a 22mer and retained high affinity ($K_d=7.3\text{nM}$) for human IL-1 α (alpha). The authors determine the crystal structure of the aptamer-IL-1 α complex at 2.1Å which leads to numerous interesting observations about the folding and structure of such modified DNA aptamers as well as potential useful structural information about IL-1 α . Finally the authors provide data demonstrating that the aptamer competes with binding of IL-1 receptor to IL-1 α and can inhibit IL-1 α induced cytokine secretion in HS27 and HuVEC-based cell assays.

Overall the results are clear and the data is both compelling and novel. Moreover the paper is very well written and describes numerous new structural motifs for modified nucleic acid aptamers. It will be of particular interest to those studying aptamers, IL-1 α and IL-1 signaling as it presents a high resolution structure of IL-1 α (albeit bound to an aptamer) for the first time. The activity data is not abundant (only cell line data and no in vivo animal data) but the paper is a beautiful structural report and not a translational research article so I believe that this amount of functional activity data is more than acceptable. Therefore I would recommend the paper for publication in Nature Communications but would ask the authors to address a few minor issues that I believe would be beneficial to the reader.

1.) It would be useful for the authors to present a space filling version of the aptamer and indicate the surface on the aptamer that is in close contact with the IL-1 α surface. This representation would be particularly useful in trying to understand how the 2Nap-dU aptamer is so compact yet interfaces with 657Å on the surface of the protein. Moreover it would be useful if the authors would speculate on if the reduced surface interaction compared to other modified DNA aptamers they have generated results in the somewhat reduced binding affinity compared to the many sub-nM binding aptamers that they have reported previously. In particular such a model would be very useful in thinking about why the aptamer is so selective for the surface of IL-1 α over IL-1 β (beta) in supplemental figure 6.

To address this point, we have included a figure showing the interface on both IL-1 α and SL1067 in Supplementary Figure 6, along with a sentence in the discussion paragraph regarding this issue. The paragraph, starting at line 382 now reads as follows.

Overall, SL1067 has a smaller interaction surface area (657 \AA^2) compared with the other three modified aptamer complexes that have been structurally characterized ($1097\text{-}1248 \text{ \AA}^2$). However, its ligand efficiency of $0.21 \text{ kcal mol}^{-1}$ per non-H contact atom is significantly larger than that of other modified aptamers ($0.12\text{-}0.18 \text{ kcal mol}^{-1}$ per non-H contact atom) (reference 21, 23). Accordingly, the calculated value of the shape complementarity (Sc) index (reference 41, 56) for IL-1 α /SL1067 is 0.725, which is at the lower end, but within the range reported for the three other modified aptamers (0.72-0.80) (reference 21). Together these parameters may indicate that the slightly reduced binding affinity of SL1067, compared with other published modified aptamers, may simply result from the reduced number of hydrophobic contact atoms (54 for SL1067 compared to 75, 78 and 131 for other published modified aptamers) (reference 23-25). They also provide evidence that in spite of the substantial hydrophobicity of the 2Nap moiety, this modification significantly contributes to the binding energy and is capable of forming a highly complementary interface with the target protein.

2.) It would be useful for the authors to speculate on why IL-1a is such a hard target and why the 2Nap-dU modification and not other similar modifications can crack this tough target case. Is there something about the "aptagenic" epitope on the surface of the protein that is only accessible to a compact aptamer (depth, width, orientation, charge, hydrophobicity etc) or are there essential conformations that 2Nap-dU modifications allow that none of the other ten SELEX experiments with seven diverse modifications could achieve?

In response to this comment we have added to the section where IL-1a as a SELEX target is discussed. We included and/or modified the following sentences starting at line 334:

While it is not obvious from a study of the structure and surface of the protein what makes IL-1 α a difficult target for SELEX, we can speculate that it may be due to a combination of factors including the small, compact nature of the protein, a low isoelectric point of 5.3 and the requirement for long-lived complexes imposed by the kinetic challenge step.

3.) In figure 4, it would be useful to indicate in panels b and c how the view is rotated relative to panel a. Otherwise it is difficult for the reader to follow.

To make this point clearer to the reader, two arrows were added to indicate the identity of the appropriate inset figure, and the degree of rotation for each inset view is now indicated.

4.) It would be useful to discuss how the key dG11 interactions are similar or different from unmodified or backbone modified aptamers as pi bonding/stacking interactions have been seen between such aptamers and their target proteins albeit not when the purines are presented in the context of a 2Nap-dU platform. Does this presentation allow dG11 adopt a conformation that allows it fit into a pocket that is otherwise not accessible?

To address this query, we have inserted the following text starting at line 221:

Although similar π bonding/stacking interactions have been seen in previous aptamer/protein complexes^{34, 35, 36}, adjacent nucleobases are not significantly involved in reinforcing the local architecture. However in the case of SL1067, the adjacent 2Nap-dU units play a special role. The 2Nap7 and 2Nap14 groups are in close proximity to dG11, where they appear to stabilize the dG11 binding pocket by making non-polar contacts with the amino acids of the pocket (Ile68 and Trp113) and resulting in a tight interaction cluster near the pocket.